# Study on Synergistic Corrosion Inhibition Effect between Calcium Lignosulfonate (CLS) and Inorganic Inhibitors on Q235 Carbon Steel in Alkaline Environment with Cl^−^

**DOI:** 10.3390/molecules25184200

**Published:** 2020-09-14

**Authors:** Bing Lin, Junlei Tang, Yingying Wang, Hu Wang, Yu Zuo

**Affiliations:** 1School of Chemistry and Chemical Engineering, Southwest Petroleum University, Chengdu 610500, China; h1900@foxmail.com (B.L.); yingyingwanglyon@126.com (Y.W.); 2School of Material Science and Engineering, Southwest Petroleum University, Chengdu 610500, China; senty78@126.com; 3Beijing Key Laboratory of Electrochemical Process and Technology for Materials, Beijing University of Chemical Technology, Beijing 100029, China

**Keywords:** corrosion inhibitor, synergistic effect, electrochemical measurements, inhibition film

## Abstract

The synergistic corrosion inhibition effect between calcium lignosulfonate (CLS) and three kinds of inorganic inhibitors (Na_2_MoO_4_, Na_2_SnO_3_, and NaWO_4_) with various molar ratios on Q235 carbon steel in alkaline solution (pH 11.5) with 0.02 mol/L NaCl was investigated by cyclic potentiodynamic polarization, electrochemical impedance spectroscopy, linear polarization, scanning electron microscopy, and X-ray photoelectron spectroscopy. Molybdate and stannate in hybrid inhibitor could promote the passivation of steel and form a complex film, which could suppress the corrosion effectively. Moreover, the insoluble metal oxides in the complex film formed by three kinds of inorganic inhibitor could help the adsorption of CLS onto the steel surface. The CLS molecules could adsorb onto the steel surface and metal oxides to form an adsorption film to protect the steel from corrosion. A three-layer protection film formed by a hybrid inhibitor, including passivation film, deposition film, and adsorption film, would effectively inhibit the corrosion reactions on the steel surface. The CLS compound with molybdate with the ratio of 2:3 shows the best inhibition effect on both general corrosion and localized corrosion.

## 1. Introduction

Alkaline environment is a common corrosion condition in reinforced concrete system, marine environment, and industrial desulfurization installation. The corrosion of carbon steel in alkaline environment has been extensively studied in recent years owing to its industrial relevance. In the reinforced concrete system, the corrosion of reinforcing steel is the main reason leading to the premature deterioration of reinforced concretes [1,2]. The corrosion of reinforcing steel in concrete is related to the presence of chloride ions and carbonation of concrete, which may result in corrosion initiation as a sequence of several events. As metal corrosion is inevitable, several techniques have been employed to reduce the rate of corrosion to a minimum, such as coatings, cathodic protection, inhibitors, and so on [3,4]. The use of corrosion inhibitors is one such strategy, which has the advantages of simple use, as well as being economical and controllable, among others. In general, corrosion inhibitors used for steel rebar in concrete include inorganic compounds, organic compounds, and hybrid inhibitors [4]. Nevertheless, traditional corrosion inhibitors for concrete have been considered to be toxic or have a poor inhibiting effect [5,6]. Hence, developing new kinds of environmental-friendly inhibitors with high inhibition efficiency for concrete rebar becomes a necessary research subject.

Owing to the increasingly prominent environmental problems, some inorganic inhibitors that are no that poisonous or are non-poisonous are receiving more and more attention. Sodium tungstate has been proven to be an environmental-friendly anodic type corrosion inhibitor for metallic matrix in a variety of corrosive media [7]. Robertson [8] first reported the inhibition effect of Na_2_WO_4_ for steel, which could form a iron-tungstate complex on the steel surface. It was also reported that tungstate lacked the oxidizing property of both chromate and nitrite [9]. Many authors published their results for the inhibiting effect of molybdate [10,11,12,13,14]. Molybdate is an environment-friendly anodic passivation inhibitor. The inhibition mechanism of molybdate has been studied by many scholars, and competitive adsorption [10], adsorption film [11,12], and oxidation [13] are the dominant views. Molybdate acts as a localized corrosion inhibitor, which is effective in the first stages of pitting initiation, and the effectiveness is decreased after the propagation stage is established [14]. As a corrosion inhibitor, Stannate could not only activate the metal surface passivated layer, but also form a protecting film on the surface of metal substrate [15,16]. Moreover, it has been proven that stannate could improve the resistance to localized corrosion [17].

However, tungstate, molybdate, and stannate are usually not used as corrosion inhibitor alone, which is owing to their low inhibition efficiency at low concentration, high cost, and low oxidizing ability. For instance, tungstate could act as an effective corrosion inhibitor for steel when the concentration exceeds 1200 ppm [18]. The effect of inhibition was obvious when the concentration of molybdate was very high (thousands of mg/L) [19]. Therefore, using co-inhibitors is useful to increase the inhibition effect of this kind of inorganic inhibitor. A combined inorganic inhibitor with organic compounds for protection of metallic matrix is supposed to have a good synergistic inhibition effect. Organic compounds with heteroatoms (P, S, N, and O) and π-bonds are the most effective inhibitors, which could adsorb on the metallic matrix by reacting with blank d-orbit of steel, and form a protective film on the matrix surface in acidic, neutral, and alkaline solution [20,21,22]. There are many investigations combining inorganic and organic inhibitors to improve the corrosion protection effect. Wu et al. [15] have studied the synergisitic inhibition effect of alkyl polyglucoside (APG) and potassium stannate (K_2_SnO_3_) for Al anode in an alkaline Al-air battery. The synergistic effect between APG and stannate can be attributed to the hydrophobic segment of APG, which could regulate the reduction of stannate in the system, leading to the more uniform deposition of Sn on the surface of the Al anode. Mehdi et al. [20] studied the synergistic behavior of sodium tungstate and penicillin G as an environment-friendly co-inhibitor on pitting corrosion for 304 stainless steel. The mixed inhibitor could provide a smooth and uniform surface owing to the formation of a protective film. It was reported [22] that the synergistic effect is present between molybdate and benzotriazole (BTA). The compounded inhibitor could promote the transformation of FeOOH and Fe_2_O_3_ in the passive film, and both passivation and pitting corrosion resistance were promoted.

As an effective corrosion inhibitor, calcium lignosulfonate (CLS) contains plenty of hydroxyl, carboxyl, and methoxyl groups, which can adsorb onto the steel surface by sharing their lone pair electron or π-electron with the free d-orbital of metal [1]. It has been proven that CLS is a good corrosion inhibitor in saturated Ca(OH)_2_ solution [23] and carbonation concrete pore solution [1] for carbon steel. The chemical structure of CLS used in this research is shown in Figure 1 [1,23]. In the preliminary work we have completed [1], the synergistic corrosion inhibition effect between CLS and sodium molybdate has been investigated and the inhibition mechanism of the hybrid inhibitor was illustrated. In order to further figure out the synergistic inhibition effect between CLS and inorganic inhibitors, the corrosion inhibition mechanism of CLS compound with different inorganic inhibitors (Na_2_MoO_4_, Na_2_SnO_3_, and NaWO_4_) was investigated by electrochemical measurements and surface analysis.

## 2. Results and Discussion

### 2.1. Electrochemical Measurements

#### 2.1.1. Cyclic Potentiodynamic Polarization (CPP) Curves

In order to understand the inhibition mechanism of CLS, molybdate, tungstate, stannate, and their compound inhibitors, potentiodynamic polarization was employed for Q235 carbon steel in pH 11.5 solution with 0.02 mol/L NaCl. Figure 2a shows the polarization curves for steel immersed in test solution with 500 ppm different inhibitors for 2 h. It can be seen that all the polarization curves show typical passivation-pitting corrosion behavior; neither organic nor inorganic inhibitor could change the corrosion behavior of carbon steel in alkaline environment. Figure 2b shows the electrochemical parameters obtained from CPP curves. The corrosion potential (E_corr_) and corrosion current density (i_corr_) were obtained using Tafel plots. The E_corr_ for steel in test solution without inhibitor is about −300 mV_SCE_, and has almost no change after adding 500 ppm CLS into the test solution, which is consistent with previous research [1]. This result indicates that CLS act as a mixed-type corrosion inhibitor [24,25]. Meanwhile, after adding 500 ppm inorganic inhibitors, the E_corr_ of steel slightly increases, and molybdate leads the largest increase of E_corr_. This result reveals that the tested inorganic inhibitors act as a mixed-type inhibitor [25,26] predominantly with anodic effectiveness [26]. The inhibition efficiency (IE%) was calculated by the following equation:(1)$$\mathrm{Inhibition}\;\mathrm{Efficiency}\;(\mathrm{IE}\%)\;=\;\;\frac{i_{\mathrm{corr}}^{0}-i_{\mathrm{corr}}}{i_{\mathrm{corr}}^{0}}\times 100$$
where i^0^_corr_ and i_corr_ represent the corrosion current density values in test solution without and with inhibitor, respectively. The IE% results shown in Figure 2b indicate that all inhibitors are moderate inhibitors for the general corrosion, and when the inhibitor concentration is 500 ppm, the IE% ranking is molybdate > tungstate > stannate > CLS.

The large positive hysteresis loops in Figure 2a are evident for all conditions, indicating that active corrosion pits initiate and propagate on the steel surface [27,28]. As pitting corrosion plays an important role on the corrosion resistance of steel exposed to chloride alkaline environment, thus CPP measurements were conducted for obtaining the typical pitting corrosion parameters [27,29]. Pitting potential (E_pit_) is the potential where the first corrosion pit appears, and 500 ppm of all kinds of inhibitors could increase the E_pit_ of carbon steel in the test environment, which means the test inhibitor could decrease the localized corrosion sensitivity of steel. The passivation region (E_pit_–E_corr_) could reflect the sensitivity of pitting corrosion [26,30]. Figure 2b indicates that 500 ppm CLS has the best inhibition ability of pitting corrosion with about a 200 mV passivation region, and the pitting inhibition ability decrease as molybdate, tungstate, and stannate. This result is consistent with the references [8,9,11], which might be owing to the different inhibition mechanism of different types of inhibitors. As an organic inhibitor, CLS could form an adsorption film on the steel surface to protect steel from localized corrosion [1,23]. In particular, CLS could be preferentially adsorbed on the active sites on a steel surface using the sinapyl alcohol group, which is beneficial for the inhibition of pitting corrosion [1,23]. The inorganic inhibitors tested in this research are an anodic passivation inhibitor, which could enhance the passivation film formed on the steel surface [9,13,17]. The protective film played a critical role on the inhibition effect of this kind inhibitor [31], and the low inhibition efficiency at a low concentration is owing to the low oxidizing ability of inhibitor. The passive current (i_pass_) means the current density limit of the passivation region where the current density remains unchanged [24,32]. Figure 2a shows clearly that i_pass_ decreases after adding 500 ppm inhibitors in the test solution, and 500 ppm molybdate treated steel have the lowest i_pass_. The re-passivation potential (E_rep_) is the intersection potential of the forward and reverse scans, and the steel potential must be above E_rep_ for existing areas of localized corrosion to propagate [33,34]. Figure 2 shows inorganic inhibitors, especially molybdate and stannate, could increase E_rep_ of carbon steel. Meanwhile, as an organic inhibitor, CLS has no effect on E_rep_.

In a word, all inhibitors tested in this research could decrease both general corrosion and localized corrosion for Q235 carbon steel in the test environment. CLS has a better effect on inhibition of the initiation of pitting corrosion, and the inorganic inhibitors could promote the re-passivation of carbon steel. The tested inhibitors have a limited corrosion inhibition effect when used alone, in order to combine different corrosion inhibition mechanisms and improve the inhibition efficiency of inhibitors, the hybrid organic and inorganic inhibitors were tested in further experiments.

Then, 500 ppm hybrid inhibitors with various ratios of CLS and inorganic inhibitor were added into the test solution to understand the synergist inhibition effect of inhibitors. The ratio of CLS to inorganic inhibitor is 400:100, 300:200, 200:300, and 100:400, respectively. Figure 3 shows the CPP curves of Q235 carbon steel in the test solution with different hybrid inhibitors, and the electrochemical parameters are shown in Figure 4. As shown in Figure 3, after adding hybrid inhibitors, the corrosion behavior of Q235 carbon steel in test solution is still passivation-pitting corrosion, and the hysteresis loops show pitting corrosion occurred on the steel surface [27,28]. Hybrid inhibitors could not change the corrosion behavior of Q235 steel in the test environment. The E_corr_ of Q235 steel in the test solution with various ratios of CLS compound molybdate or tungstate decreases obviously, and as the inorganic inhibitor ratio decreases, the E_corr_ slightly decreases. This result is consistent with the literature [1,35]. Meanwhile, the E_corr_ of steel in solution adding CLS and stannate shows almost no change. Figure 4b shows the i_corr_ and inhibition efficiency (IE%) of hybrid inhibitors. All hybrid inhibitors could decrease the i_corr_ of steel in the test solution, and the IE% value is obviously higher than that when using any single tested inhibitor. When a certain ratio hybrid inhibitor is added into the test solution, the inhibitor compounded by CLS and molybdate has the highest inhibition efficiency, and the inhibitor containing stannate has the lowest. This result is consistent with IE% in Figure 2b, which might be related to the difference in inhibition ability between inorganic inhibitors. The ratio of inorganic inhibitor in hybrid inhibitors plays an important role on inhibition efficiency; as the inorganic inhibitor ratio increases to 2:3, the IE% increases to the highest value, and then slightly decreases. CLS and inorganic hybrid inhibitors have a synergistic inhibition effect, which could suppress the general corrosion of carbon steel in the test solution, and the best ratio of organic and inorganic inhibitor is 2:3.

For the localized corrosion, as shown in Figure 3 and Figure 4, the E_pit_ of Q235 steel in solution with compound inhibitors increased obviously, especially for the CLS compounded with molybdate and stannate. The E_pit_ slightly increases as the inorganic inhibitor ratio increases, which is consistent with the previous research [1]. The E_pit_–E_corr_ value of CLS compounded with molybdate is highest in all tested hybrid inhibitors, which means CLS and molybdate hybrid inhibitor have the best effect on decreasing susceptibility of pitting corrosion. The increasing E_pit_ and passivation region of Q235 steel in alkaline solution with hybrid inhibitor are owing to the effect of reducing the number and magnitude of metastable pitting transients [29,30], which means the hybrid inhibitor could block the active sites on the steel surface and affect the nucleation of pits [1], leading to a decrease of pitting corrosion susceptibility [36]. E_rep_ of Q235 steel in test solution increases obviously, especially for the CLS compounded with molybdate or tungstate, while the hybrid inhibitor with stannate has little effect on increasing E_rep_. This result indicates that a high ratio of molybdate and tungstate in the hybrid inhibitor could promote the re-passivation of the Q235 steel surface. It has been reported that molybdate or tungstate could not be inhibited or might even be promoted if their concentration is too low [36,37]. The increase of E_rep_ indicates the increase of the re-passivation ability of carbon steel. When pits appear on the steel surface, the pH value in the corrosion pit environment is decreasing, which will lead to the concentration gradients for mass transport and promote further pit nucleation [38]. It is well known that, in the low pH environment, MO_4_^2−^ and WO_4_^2−^ condense into various polymolybdate and polytungstate [7,11], respectively. However, the structure of polytungate is more complicated than that of polymolybdate [11]. Polyanions have a chelate effect with Fe^3+^, which could form quickly and react with the blank orbit(d) of steel to form complexes, and then the complexes could adsorb onto the pit bottom to inhibit the growth of pitting corrosion [7]. The hybrid inhibitor could help re-passivation of the pit, and the inhibition effect is much better than using any tested inorganic inhibitor alone. This result is because the adsorption film formed on the steel surface could reduce the local acidification process in the pits. The i_max_ shown in Figure 3 further confirmed the results of hybrid inhibitor could increase the re-passivation ability of carbon steel. I_max_ is related with anodic dissolution of steel, and indicated the maximum current density during the pitting process [24,26,39]. As the inorganic inhibitor ratio increased, i_max_ decreased obviously, which means the pits appearing on the steel surface have been inhibited.

CLS compound with three kinds of inorganic inhibitors could inhibit both general corrosion and localized corrosion. CLS compound with molybdate has the best inhibition effect, while CLS compound with tungstate has a better effect on localized corrosion, and compound with stannate has a better effect on general corrosion. The inhibition effect of the hybrid inhibitor increases as the inorganic ratio increases, and the inhibition effect for both general corrosion and localized corrosion is best when the CLS and inorganic inhibitor ratio is 2:3. As the inorganic inhibitor ratio further increases, the inhibition effect slightly decreases.

#### 2.1.2. Electrochemical Impedance Spectroscopy Measurement

In order to further understand the inhibitive behavior of compound inhibitors, a series of electrochemical impedance spectroscopy (EIS) tests were carried out in test solution with 500 ppm various ratios of different compound inhibitors. The Nyquist and Bode plots for steel samples immersed in test solution with hybrid inhibitors for 2 h are presented in Figure 5. As seen from the Nyquist plots, a single depressed capacitive semicircle is presented in all the Nyquist plots, and the capacitive semicircle radius increases obviously after adding inhibitors in the test solution. This result reveals that the addition of hybrid inhibitor did not change the mechanism of the corrosion process, but inhibited corrosion by forming a protective film on the steel surface [40,41], which is consistent with the results of polarization curves. For each kind of hybrid inhibitor, as the inorganic inhibitor ratio increases, the capacitive semicircle radius of Nyquist plots decreases obviously. This result indicates that the shielding effectiveness of the protective film formed on the steel surface decreases as the inorganic inhibitor ratio increases, and the shielding effectiveness of inhibitor film mainly depends on the ratio of CLS in compounds [1,15,20]. Moreover, the absolute value of the impedance decreases as the inorganic inhibitor ratio increases for each kind of hybrid inhibitor in Bode plots at a low frequency (0.01 Hz). Simultaneously, phase angle plots were observed to follow a similar trend as the Nyquist and Bode plots. The decrease of CLS concentration leads to the surface coverage by the adsorbed inhibitor film [40].

Figure 6 reveals the equivalent circuit, which could describe the solution/steel interface. In the equivalent circuit, R_s_ stands for the solution resistance and R_ct_ represents the charge-transfer resistance corresponding to the corrosion reaction at the metal substrate/solution interface. C_dl_ is replaced by constant phase elements (CPEs), which are used in the model to compensate for the non-homogeneous electrode surface [15,41,42]. R_film_ and C_film_ represent the resistance and capacitance, respectively, of the protective film formed on Q235 steel surface, which includes passivation film, precipitation film, and adsorption film. The fitted parameters of EIS measurement are shown in Figure 7.

From the fitted parameters presented in Figure 7, the R_ct_ increases as the inhibitor is added into test solution, which suggests the hybrid inhibitor could inhibit the corrosion reaction in the steel/solution interface. As the inorganic inhibitor ratio increases, the R_ct_ increases, and the maximum R_ct_ value is reached when CLS/inorganic ratio is 2:3, and then decreases slightly. The increasing R_ct_ suggests the corrosion reaction was inhibited by the synergistic effect of CLS and inorganic inhibitor, and the most efficient hybrid inhibitor ratio might be 2:3, which is consistent with the CPP results. R_film_ shows a slight decrease as the inorganic inhibitor ratio increases. The R_film_ consists of the passivation film, precipitate film, and adsorption film. As the CLS concentration decreases, the barrier effect of adsorption film formed by organic inhibitor decreases, which is dominant in R_ct_ value [15,18]. For three kinds of hybrid inhibitors, CLS compound with molybdate has the highest R_ct_ and R_film_, which means molybdate has the best synergistic inhibition effect with CLS. Moreover, stannate or tungstate have almost the same synergistic effect with CLS, which could confirm the results of CPP curves. The C_dl_ and C_film_ increases as the inorganic inhibitor ratio increases, which suggests that states of steel/solution interface have been changed by the hybrid inhibitor. Although the barrier effect of CLS adsorption film decreases, the double-layer protection film formed by passivation film and adsorption film could further protect the steel surface [23]. The C_dl_ and C_film_ for the steel treated by CLS and molybdate suggest that this hybrid inhibitor has the best synergistic inhibition effect. The C_dl_ of CLS and stannate or tungstate has almost no difference, while the C_film_ of CLS compound with tungstate treated steel is much lower than that treated by CLS and stannate.

From the EIS resluts, molybdate has the best synergistic inhibition effect with CLS, followed by tungstate and stannate, and the best ratio of CLS and inorganic inhibitor is 2:3, which is consistent with the results of CPP curves. The hybrid inhibitors could increase R_ct_ and R_film_, and lead to the decrease of C_dl_ and C_film_ by promoting the formation of a protection film on the steel surface, which is formed by passivation film and adsorption film.

#### 2.1.3. Effect of Pre-Filming Time on Compound Inhibitors

The pre-filming time has an important role in the inhibition processes for the inhibitor [43,44]. In order to understand the effect of pre-filming time on different hybrid inhibitors, 500 ppm hybrid inhibitor with the best ratio, 200 ppm CLS and 300 ppm inorganic inhibitor, was added into the test solution. Immersion tests for Q235 steel in solution without inhibitor and with 500 ppm CLS are performed as comparative experiments. As shown in Figure 8a, the E_OCP_ of Q235 carbon steel immersed in test solution without inhibitor decreases from about −300 mV (saturated calomel electrode, SCE) to −700 mV (SCE) as the immersion time increases. After 24 h immersion, the open circuit potential (OCP) of steel is going to stabilize at about −700 mV (SCE), which indicates the relative close rates of the corrosion process and formation of oxides at the interface [42]. The OCP of Q235 carbon steel immersed in test solution with 500 ppm CLS for different immersion time is about −290 mV (SCE), and shows a slight decrease as the immersion time increases. This result is consistent with Figure 2, indicating CLS acts as a mixed-type inhibitor. As an organic inhibitor, CLS could form an adsorption film on a steel surface after 10 h pre-film [1], which could reduce the content of Cl^−^ adsorbed on the steel surface and prevent the local enrichment of Cl^−^ [23]. For the steel immersed in the test solution with compound inhibitors, the OCP for the first hour is about −300 mV (SCE), and slightly decreases in the first 4 h; as the immersion time further increases, the OCP slightly increases. After 48 h immersion, the compound inhibitor with molybdate shows the highest OCP value, followed by the compound inhibitor with NaWO_4_ or Na_2_SnO_3_. Similar to the CPP curves results, compound inhibitors act as a mix-type inhibitor. The decreases of OCP in the first 4 h might be owing to the adsorption of CLS onto the steel surface, which is much shorter than adding CLS alone and suggesting the inorganic inhibitor could promote the adsorption of CLS [15,18]. The increases of OCP from 4 h to the end of test are owing to the oxidation capacity of inorganic inhibitor [11,14,15], and the passivation film and deposition film formed on the steel surface could suppress the corrosion reactions. The OCP value after 48 h for different hybrid inhibitors used to treat the steel surface confirmed these results.

Figure 8b shows the R_p_ values and the IE% calculated by R_p_ of Q235 carbon steel immersed test solution without and with 500 ppm different inhibitors. It can be seen that, as the immersion time increasing, the R_p_ of steel in solution without inhibitor decreases slightly, which is owing to the corrosion reactions on the steel surface. After adding 500 ppm inhibitor, the R_p_ value shows an increasing tendency, which is owing to the protection film formed on the steel/solution interface. The protection film formed on the steel surface has a barrier effect to repress the corrosion reactions [42,45]. As the immersion time increases, the R_p_ value of steel treated by inhibitor remains at a high level, which means that, in 48 h, the inhibitors have a sufficient protection effect. The CLS compound with molybadte treated steel has the highest R_p_, followed by the CLS compound with stannate or tungstate. According to the inhibition efficiency (IE%) calculated from the R_p_, molybdate has the best synergistic inhibition effect with CLS, followed by tungstate and stannate. This result confirmed the results of the CPP curves.

### 2.2. Surface Analysis

#### 2.2.1. Scanning Electron Microscope

Scanning electron microscope (SEM) was carried out to investigate the surface morphology and elements’ distribution of Q235 coupons immersed in the solutions with difference hybrid inhibitors for 24 h. According to the SEM images shown in Figure 9, the Q235 steel demonstrates very different surface conditions. For the steel immersed in solution without inhibitor, there are plate-shaped corrosion pits. This might be owing to the pH value (11.5) and Cl^−^ concentration of the test solution, as the passivation film formed on the steel surface could not provide enough protection for steel [2]. There are almost no corrosion marks on the steel surface treated by hybrid inhibitors. For the CLS and molybdate treated steel surface, there are dendritic adsorbents. On the CLS and stannate or tungstate treated steel surface, the polish marks are clear, and there are depositions on the steel surface.

In order to further understand the surface morphology and elements’ distribution, energy dispersive X-ray spectrometer (EDS) was carried out on SEM figures with high resolution, and the results are shown in Figure 10. For the steel treated with CLS and molybdate, there are dense attachments constituting the adsorption film, and the protection film could cover the grooves formed by the polishing step. From the EDS results, there is Fe element attribution to metal substrate or Fe oxides, Mo element attribution to MoO_4_^2−^ or MoO_2_ deposition on the steel surface, and Ca deposition and S element represent the CLS adsorption film. For the CLS compound with stannate or tungstate treated steel surface, the deposition area is much smaller than that of the CLS compound with molybdate. From the EDS results, the elements on the hybrid inhibitor treated steel surface are similar to the surface treated by the hybrid inhibitor containing Mo.

#### 2.2.2. X-ray Photoelectron Spectroscopy Analysis

In order to further understand the distribution of chemical elements on the Q235 carbon steel surface treated by compound inhibitors, X-ray photoelectron spectroscopy (XPS) measurement was performed on the surface of samples immersed in the test solution with 500 ppm different compound inhibitors for 24 h. The high resolution spectra of elements on steel surface are shown in Figure 11, Figure 12 and Figure 13.

Figure 11 shows the high resolution spectra of C 1s, Ca 2p, and S 2p on the steel surface, which are all the characteristics elements of the CLS molecule. The C 1s spectra represented in Figure 11a shows four peaks. The first peak at 284.8 eV is attributed to the C-C/C-H bonds [28,46] in the CLS molecules. The second peak at 286.8 eV for adding CLS along can be attributed to the carbon atoms bonded to S in C-S bonds [46], which shifted to more negative when adding hybrid inhibitors. The decreases of C-S bonding energy indicates that the S atoms in C-S bonds accept the feedback electrons from Fe atoms [21,46], which means the inorganic component in the hybrid inhibitor could promote the adsorption of CLS onto the steel surface [1,22]. The third peak at 288.3 eV is ascribed to C-O bonds [47,48], and the last peak at 292.7 eV is assigned to the benzene ring [48] in the CLS molecules. The C 1s spectra is consistent with the chemical structure of the CLS molecule and the addition of inorganic inhibitors has no influence on the chemical structure of LS^−^, and confirms the adsorption film formed by CLS on steel surface.

Figure 11b shows the Ca 2p and S 2p spectra on CLS or the hybrid inhibitor treated steel surface. As characteristic elements of CLS, Ca and S elements on steel surface could further confirm the adsorption film formed by CLS. The Ca 2p spectra are decomposed into two peaks at 347.1 eV and 350.8 eV for all conditions. The first peak at 347.1 eV is attributed to CaO/Ca(OH)_2_ [22,49]. The hydrolysis of CLS in the alkaline solution could generate Ca^2+^ and LS^−^, and CaO/Ca(OH)_2_ precipitation on the steel surface could cover the steel surface and prevent the corrosion [22,46,50]. The second peak at 350.8 eV is attributed to the Ca atoms bonding with LS^2−^ by Ca-O-S bonds [22] on the steel surface, and Ca^2+^ could promote the adsorption of the organic inhibitor onto the steel surface. There is a synergistic corrosion inhibition effect between Ca^2+^ and LS^−^. The addition of inorganic inhibitors has almost no influence on the adsorption and precipitation of Ca^2+^. The S 2p spectra show two peaks on the CLS treated steel surface; the peaks at 168.4 and 167.0 eV [22,47] represent S atoms in Ca-O-S bonds and LS^2−^, respectively. The first peak indicates the co-adsorption of LS^2−^ and Ca^2+^, and the second one represents the LS^−^ direct adsorption on the steel surface. For the hybrid inhibitor treated steel surface, there is only one widened peak at about 168.0 eV, which might be owing to the metal-S-O bonds, where metal represents Ca^2+^; Fe^2+^/Fe^3+^; and Mo, Sn, W in the inorganic inhibitor. This result confirmed that the inorganic inhibitor could promote the adsorption of the organic inhibitor.

Figure 12 shows the high-resolution Mo 3d, Sn 3d, and W 4f spectra detected on the steel surface treated by 500 ppm inorganic inhibitor or hybrid inhibitor. There are two peaks for Mo 3d spectra; the peak at 231.9 eV is typical of Mo^4+^ (such as MoO_2_) [51], which could confirm the reduction of molybdate. The reduction of Mo^6+^ could help to form a more stable passivation film on the steel surface [52,53]. Furthermore, the peak at 235.1 eV is related to Mo^6+^ [22], which may be associated with MoO_4_^2−^ ions ((metal)_x_-(MoO_4_)_y_) adsorption film. There is a precipitation film formed above the passivation film by MoO_x_, Fe_x_O_y_, and metal molybdate salt. There are two main peaks shown in Figure 12b at 495.2 eV and 486.6 eV corresponding to Sn 3d_5/2_ and Sn 3d_3/2_ states, respectively, with a binding energy separation of 8.6 eV from which the oxidation state of Sn is inferred to be as 4^+^ states [54,55]. The peaks at 496.1 eV and 487.5 eV might be associated with Sn^2+^ states [55], which is the reduction of the stannate. The negative shift from 496.7 eV for stannate treated steel to 496.1 eV for the hybrid inhibitor treated steel surface is owing to the Sn^2+^ accepting the electrons from CLS molecules [21,46]. This result indicates the Sn_x_O_y_ deposited on the steel surface could enhance the adsorption of the organic inhibitor. For the stannate treated steel, the Sn^2+^ area is much larger than that of the hybrid inhibitor treated steel, which means there is still sufficient precipitation of statnnate on the steel surface to protect the surface from corrosion. The high resolution spectrum of W (Figure 12c) shows peaks at 37.4 eV and 35.3 eV, corresponding to W 4f_7/2_ and W 4f_5/2_ in tungstate, respectively, which are consistent with W^6+^ in the inorganic inhibitor [56,57]. The negative peak shift is because the tungstate adsorbed on the steel surface could enhance the adsorption of CLS molecules. While no low-balance W could be found in the spectra of steel treated by tungstate or hybrid inhibitor containing W, these results indicate that the tungstate could not enhance the passivation reaction in the test environment.

The O 1s and Fe 2p_3/2_ peaks for Q235 steel treated by 500 ppm CLS or hybrid inhibitors are shown in Figure 13. The peak at 530.1 eV corresponds to O^2−^, which is attributed to the O atoms bonds with metal in Fe_2_O_3_ and Fe_3_O_4_ [22,28]. The peak at 528.7 eV is attributed to the O^2−^ bonded with Mo, Sn, and W atoms in hybrid inhibitors, which is because these inorganic atoms have more unoccupied orbitals to bond with oxygen [53]. The peak at 531.6 eV can be attributed to the metal oxides and hydroxides or H_2_O species on the steel surface [28,58]. This peak area increases obviously after adding the hybrid inhibitor, which might be because the inorganic inhibitor could promote the metal oxides and hydroxides through oxide reactions to the Fe matrix. The peak may be assigned to the oxygen atoms of the O-C bond in the organic compounds from surface contamination and from the adsorbed water at 533.7 eV [46,51]. This peak for CLS treated steel is much larger than the hybrid inhibitor treated steel, which means this peak is related to the organic inhibitor concentration.

The deconvolution of the high-resolution Fe 2p3/2 spectra of the Q235 steel surface treated by 500 ppm CLS or hybrid inhibitor has four peaks. The first component at 705.8 eV is attributed to metallic iron (Fe^0^) [28]. The peaks at 707.2 eV and 710 eV can be attributed to the Fe^2+^ present in FeO [28,29] and Fe_3_O_4_ [46], respectively. The peak at 712.4 eV is assigned to Fe^3+^, which can be associated with the ferric oxide/hydroxide species such as Fe_2_O_3_, Fe_3_O_4_, and FeOOH [46,47]. The ratio of different components of Fe in the inhibitor treated steel surface is listed in Table 1. The ratio of Fe^3+^/Fe^2+^ could be used to evaluate the stability of the passivation film on the steel surface [29]. For the steel immersed in the test solution with 500 ppm CLS, the Fe^2+^ ratio is the highest. This result might be owing to the adsorption of CLS, replacing water molecules on the steel surface and reducing the passivation tendency of steel. For the steel treated by the hybrid inhibitor, the ratio of Fe^3+^/Fe^2+^ decreases with Mo, Sn, and W, which is related to the oxidation ability of inorganic inhibitor. In addition, Sn treated steel surface has the lowest ratio of Fe^2+^, which might be because the molybdate could also react with Fe^2+^ to form a protective film.

## 3. Discussion

### 3.1. Corrosion Inhibition Mechanism of Tested Inhibitors

In alkaline environment with chlorides, the passivation film formed on steel surface could increase the corrosion resistance. Because to the passivation film formed on the steel surface is always homogeneous, some weak regions could be destroyed by Cl^−^ and result in the formation of corrosion pits. Therefore, the damage regions on passivation film acts as an anode, whereas the complete passivation film serves as a cathode. Accordingly, the electrochemical reactions that occurred on the steel surface when exposed to alkaline solutions are as follows [25,59]:Anodic reaction: Fe→Fe^2+^ + 2e^−^(2)
Cathode reaction: O_2_ + 2H_2_O + 4e^−^→4OH^−^(3)
Corrosion products formation: Fe^2+^ + 2OH^−^→Fe(OH)_2_→Fe(OH)_3_(4)

As an organic inhibitor, CLS molecules could dissociate instantly to lignosulfonate ions (LS^2−^) and calcium ions (Ca^2+^). The steel surface in alkaline solution has extra positive charges [23], which could provide a driving force for the adsorption of negative ions from solution [24,60]. The competitive adsorption between lignosulfonate ions and Cl^−^ could reduce the corrosion reaction and protect the steel surface. The benzene rings and sulfo-group in CLS molecule could share the π-electron or lone pair electron with free d-orbital of the steel surface and form a stable adsorption film [1,23,45], where the film protects the steel by a physical shielding effect. The adsorption film formed by inhibitor and passivation film formed in alkaline environment form a double-layer barrier on the steel surface and prevent the dissolution of metal [1,61]. At the same time, Ca^2+^ in alkaline environment can form Ca(OH)_2_ or CaO, which could precipitate on the steel surface and promote the adsorption of LS^2−^ by founding Ca-S-O bonds [1,23].

Sodium molybdate is an environment-friendly anodic passivation inhibitor [62], and MoO_4_^2−^ could enhance the passivation film of carbon steel. Molybdate could reduce the general corrosion and suppress the propagation of corrosion pits. In addition, molybdate can also react with Fe^2+^ and Fe^3+^ to from a protective film by the following equation:Fe^2+^ + MoO_4_^2−^→FeMoO_4_(5)
2Fe^3+^ + 3MoO_4_^2−^→Fe_3_(MoO_4_)_2_(6)

A precipitation film composed of MoO_2_, FeMoO_4_, and Fe_3_(MoO_4_)_2_ forms on the steel surface as a physical barrier, which can be confirmed by the EDS and XPS results.

Sodium tungstate is an environment-friendly anodic type corrosion inhibitor [20], while the oxidation capacity is weaker than molybdate [11,14,63]. The passivation film formed in solution containing tungstate could suppress the nucleation and propagation of corrosion pits [6], while from the XPS results, tungstate has little effect on enhancing the passivation film in the test environment. Tungstate could also react with Fe^2+^ to form FeWO_4_ and form a protective film. As a corrosion inhibitor, sodium stannate could not only active the surface passivated layer, but also form a protecting film on the surface of metal during the reaction process [15]. However, the improvement of corrosion inhibition efficiency is limited owing to the weak oxidation capacity and uneven deposition of the protecting film [15,64]. There is research [17] pointing out that Na_2_SnO_3_ has an inhibition effect only on the outside of the pits.

The three kinds of inorganic inhibitors investigated in the research have oxidation capacity to promote the passivation form on the steel surface, and the oxidation capacity decreases with molybadte, stannate, and tungstate. Moreover, these inhibitors could also form a precipitation film to protect the steel surface.

### 3.2. Synergistc Inhibition Effect between CLS and Inorganic Inhibitors

The inhibition mechanism for the hybrid inhibitors could be inferred using the above results, and the illustration of the passivation–adsorption structure is shown in Figure 14. The first step is for the hybrid inhibitor to dissociate instantly to lignosulfonate (LS^2−^), calciumions (Ca^2+^), and MoO_4_^2−^/WO_4_^−^/SnO_3_^2−^. The negative inhibitor ions would adsorb on the steel surface competitively against Cl^-^ to protect the passive film.

The adsorbed molybdate or stannate would promote the passivation of steel surface to form Fe oxides and the reduction products of inorganic inhibitors. A deposition film formed by inorganic inhibitors on passivation film could further protect the steel surface, which could also enhance the adsorption of CLS by forming metal-O-S bonds [22] to form a three-layer protection system. For the hybrid inhibitor with tungstate, it could react with Fe^2+/3+^ to form a deposition film. At the same time, the Ca^2+^ ions in alkaline solution can form Ca(OH)_2_ and further form CaO, which could precipitate on the steel surface to enhance the deposition film. In addition, the inorganic inhibitors could react with Ca^2+^ to form an insoluble precipitate. Finally, LS^2−^ would adsorb on the outside film by sharing electrons to metal atoms, forming a hydrophobic layer to protect the steel.

The three-layer protection system formed by the hybrid inhibitor could prevent both general corrosion and localized corrosion. Once pitting corrosion occurs on the steel surface, the hybrid inhibitor could also enhance the re-passivation process once the pitting corrosion formed on steel surface, especially for the inhibitor compound with molybdate or tungstate. Under the acid environment at the pit bottom, polytungstate ions can form easily via the following reactions [7,11]:H^+^ + WO_4_^2−^→[HW_6_O_20_]^3−^→[H_2_W_12_O_40_]^6−^(7)
WO_4_^2−^→[W_7_O_24_]^6−^→[W_10_O_32_]^4−^→[W_6_O_19_]^2−^(8)
2H^+^ + 6WO_4_^2−^ + 12[W_7_O_24_]^6−^→7[W_12_O_42_H_2_]^6−^(9)
7H^+^ + 6WO_4_^2−^→HW_6_O_21_^5−^ + 3H_2_O(10)

Molybdate ions could form polymolybdate ions by the following reactions [11]:7MoO_4_^2−^ + 8H^+^→[Mo_7_O_24_]^6−^ + 4H_2_O (11)
7[Mo_7_O_24_]^6−^ + 20H^+^→Mo_8_O_26_^4−^ + 4H_2_O(12)
Polytungstate ions can form quickly and react with the blank orbit (d) of steel to form complexes, and then adsorb onto the steel surface to form a dense adsorption layer [65] in corrosion pits to inhibit the further growth of pitting corrosion.

## 4. Experiment Details

### 4.1. Materials and Test Solutions

Q235 carbon steel was studied in this research with the following chemical composition (wt%): C 0.13, Si 0.29, Mn 0.76, S 0.11, Cu 0.08, and Fe balance. The test sample size was 8 mm × 8 mm × 10 mm, which was set in epoxy resin, leaving a 0.64 mm^2^ area exposed to the test solution. The working surface was prepared and abrade by emery paper from 240 to 1000 grade, then washed with de-ionized water and ethanol, and finally dried by hot air.

The corrosion medium was composed of 0.0021 mol/L NaOH, 0.0042 mol/L KOH, and 0.02 mol/L NaCl, and the pH value was adjusted to 11.5 by NaOH or HCl, which was used to simulate the pore solution for a carbonated concrete environment [66]. Calcium lignosulfonate, sodium molybdate, sodium tungstate, and sodium stannate were added to test solution as corrosion inhibitors. All chemical medicines used in this research are of analytical grand, and all tests were carried out at room temperature (25 °C).

### 4.2. Electrochemical Measurements

All electrochemical tests were performed in a standard three-electrode cell using a CS350 electrochemical workstation (Corrtest Company, Wuhan, China). Q235 carbons steel with a certain area of 0.64 mm^2^, saturated calomel electrode (SCE), and platinum electrode were used as the working electrode, reference electrode, and counter electrode, respectively. Four parallel tests were run under each experimental condition.

After immerging the Q235 steel samples in test solutions for 2 h, the cyclic potentiodynamic polarization (CPP) curves were started at a potential 300 mV below the open circuit potential (OCP) at a scan rate of 0.1 mV/s in anodic direction until the current density increased up to 0.02 mA/cm^2^, then the scanning direction reversed at the same scanning rate until the test ended. The electrochemical impedance spectroscopy (EIS) was performed under OCP with a potential range from 0.01 Hz to 100 kHz, with a potential perturbation of 30 mV. The impedance data were fitted by ZSimpWin software.

OCP and linear polarization tests were employed to investigate the effect of pre-filming time on inhibition effect. The OCP measurement was conducted to record the variation of the potential for the steel coupons immersed in test solution without or with inhibitor for a certain time. Linear polarization tests, which were carried out for each sample after the OCP test, were used to measure the polarization resistance (R_p_) by potential scanning at the rate of 0.1667 mV/s under a potential range from −10 mV to +10 mV (vs. OCP), and the results were fitted by CVIEW2 software.

### 4.3. Surface Analysis

The surface morphology and chemical composition of inhibitor treated for 24 h carbon steel surface were observed by a scanning electron microscope (SEM) equipped with an energy dispersive X-ray spectrometer (EDS). The SEM observation was performed by a Quanta 650 instrument (FEI, Hillsboro, OR, USA).

An X-ray photoelectron spectroscopy (XPS) test was performed on inhibitor treated steel surface in order to further confirm the surface composition with a Thermo Fisher ESCALAB 250 spectrometer (Qaltham, MA, USA). The binding energy values were calibrated using the C 1s peak at 248.8 eV.

## 5. Conclusions

The synergistic inhibition effect and mechanism of CLS compound with molybdate, stannate, and tungstate in alkaline environment with 0.02 mol/L Cl^−^ were studied in this work, and the following conclusions can be drawn:

The electrochemical measurements including CPP curves and EIS results reveal that hybrid inhibitors have a better inhibition effect than using CLS or inorganic alone at the same concentration for both general corrosion and localized corrosion. CLS has a synergistic inhibition effect with molybdate, tungstate, and stannate for carbon steel in the test alkaline environment by increasing the pitting potential and decreasing the corrosion current density. The hybrid inhibitor formed by CLS and molybdate shows the best inhibition effect out of all hybrid inhibitors, and the best ratio of CLS and inorganic inhibitor is 2:3. The E_ocp_ and R_p_ test for different immersion time indicate that the hybrid inhibitor has an inhibition effect for the whole test time.

The electrochemical measurements and surface analysis confirmed the protection film formed on the steel surface and the inhibition mechanism for the hybrid inhibitors could be inferred. The inorganic inhibitor (molybdate and stannate) could enhance the passive film on the steel surface. Oxides and hydroxides of Ca, Mo/Sn, and insoluble inorganic inhibitor compounds could deposit on the steel surface, forming a deposition film. Finally, CLS would adsorb on the outside of the steel surface, forming hydrophobic layer to protect the steel. The three-layer compounded film effectively inhibits corrosion of the steel surface.

## Figures and Tables

**Figure 1 molecules-25-04200-f001:**
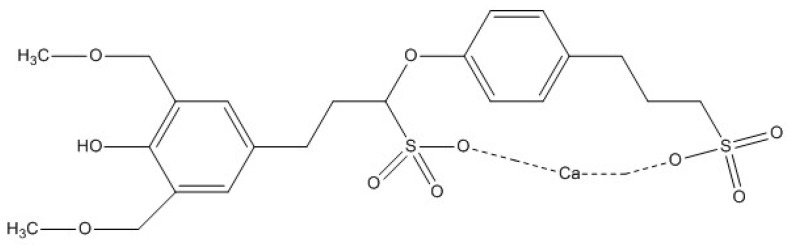
Chemical structure of calcium lignosulfonate (CLS) [1,23].

**Figure 2 molecules-25-04200-f002:**
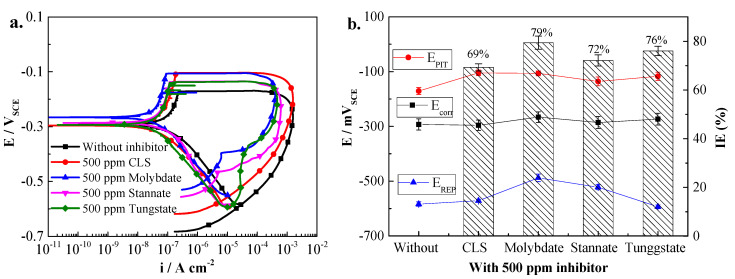
Cyclic potentiodynamic polarization (CPP) curves of Q235 carbon steel in test solution with different inhibitors (**a**) and electrochemical parameters of CPP curves (**b**). CLS, calcium lignosulfonate; IE, inhibition efficiency.

**Figure 3 molecules-25-04200-f003:**
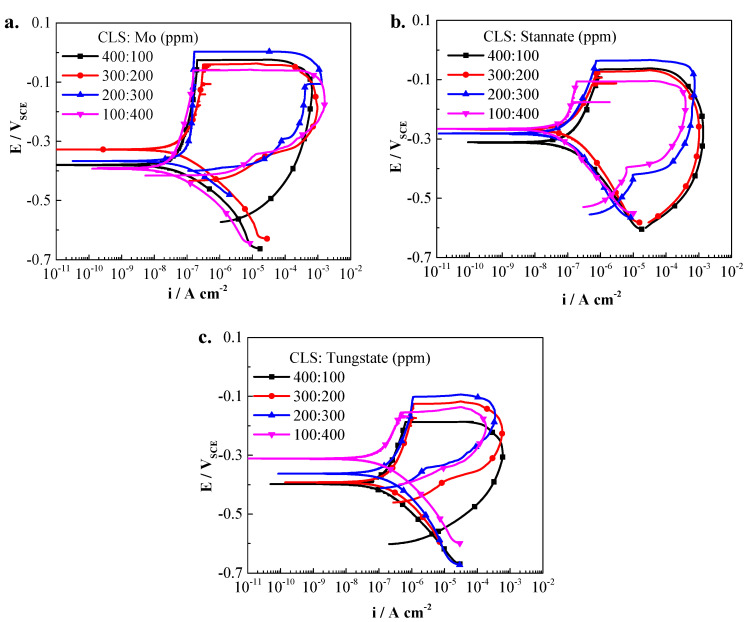
Polarization curves of Q235 steel in solution with CLS and inorganic inhibitors. (**a**) CLS with molybadte; (**b**) CLS with stannate; (**c**) CLS with tungstate.

**Figure 4 molecules-25-04200-f004:**
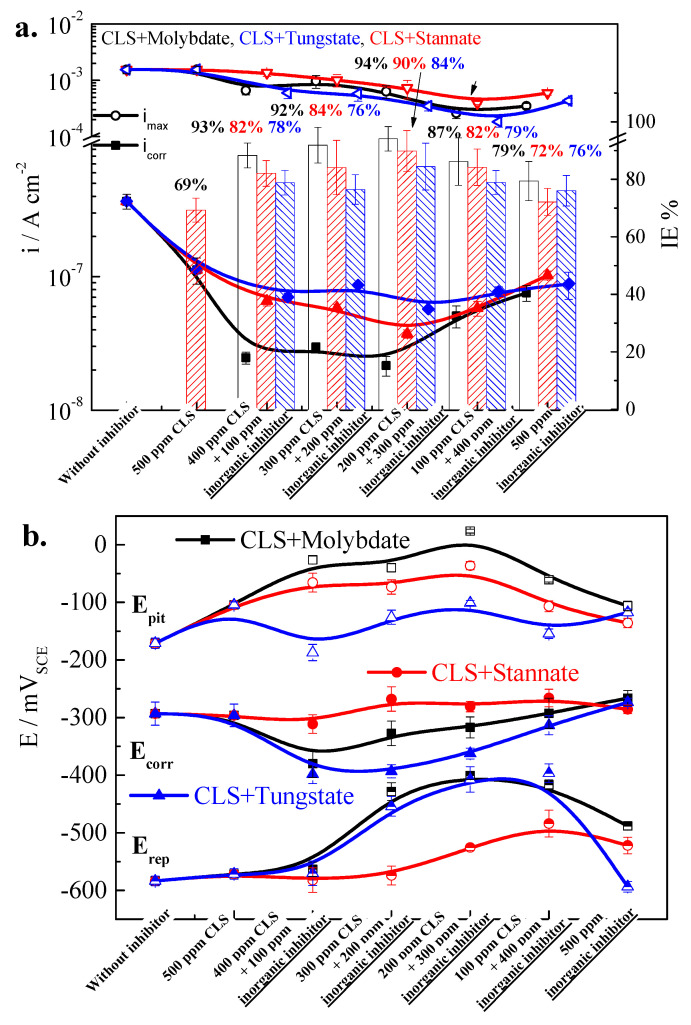
Electrochemical parameters of CPP curves for compound inhibitors: (**a**) E_ocp_, E_pit_, and E_rep_; (**b**) i_corr_, i_max_, and IE%.

**Figure 5 molecules-25-04200-f005:**
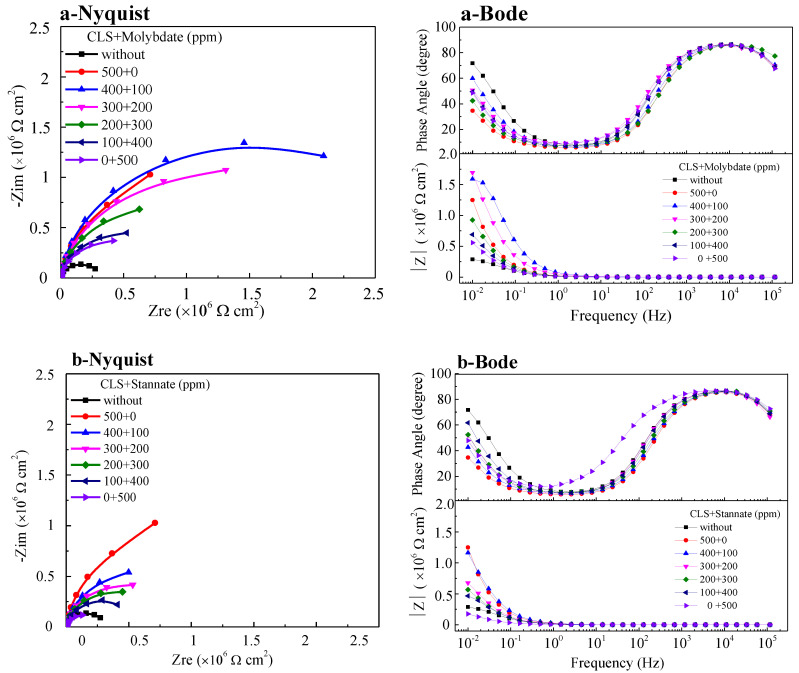
Nyquist plots and Bode plots of Q235 carbon steel in test solution with different compound inhibitors: (**a**) CLS with molybdate; (**b**) CLS with stannate; (**c**) CLS with tungstate.

**Figure 6 molecules-25-04200-f006:**
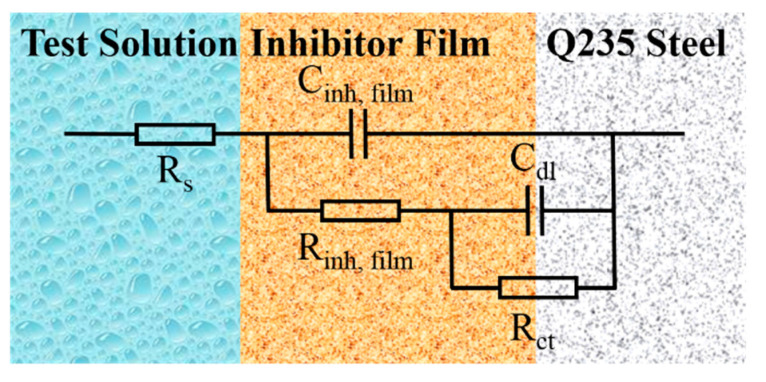
The equivalent circuit for electrochemical impedance spectroscopy (EIS) measurement.

**Figure 7 molecules-25-04200-f007:**
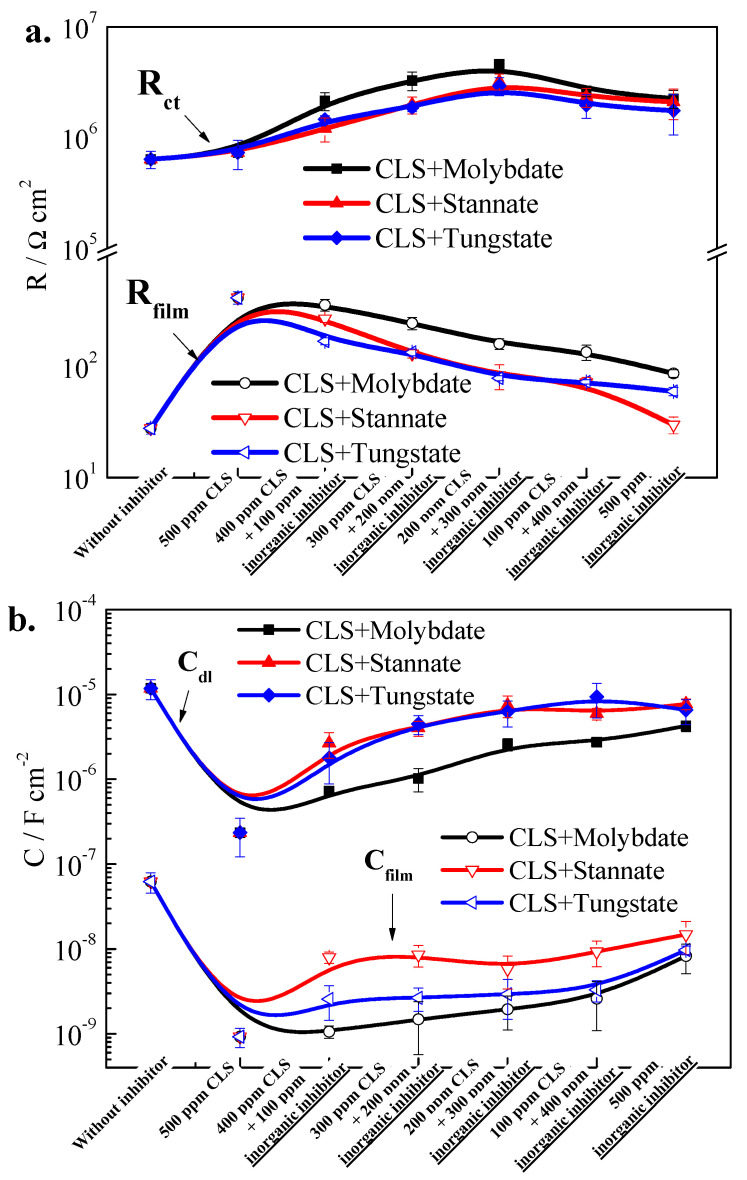
Electrochemical parameters of EIS; (**a**) R_ct_ and R_film_; (**b**) C_dl_ and C_film_.

**Figure 8 molecules-25-04200-f008:**
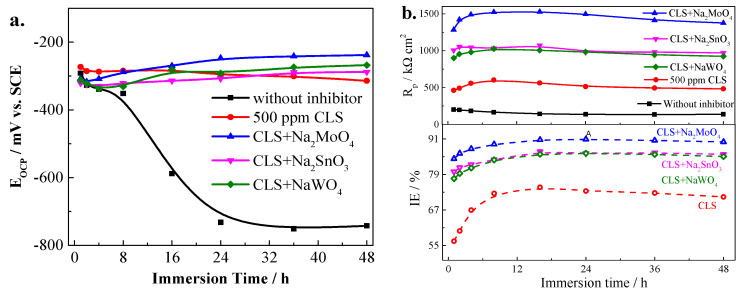
E_OCP_ (**a**) and R_p_ (**b**) values of Q235 carbon steel in alkaline test solution with different compound inhibitors. SCE, saturated calomel electrode; OCP, open circuit potential.

**Figure 9 molecules-25-04200-f009:**
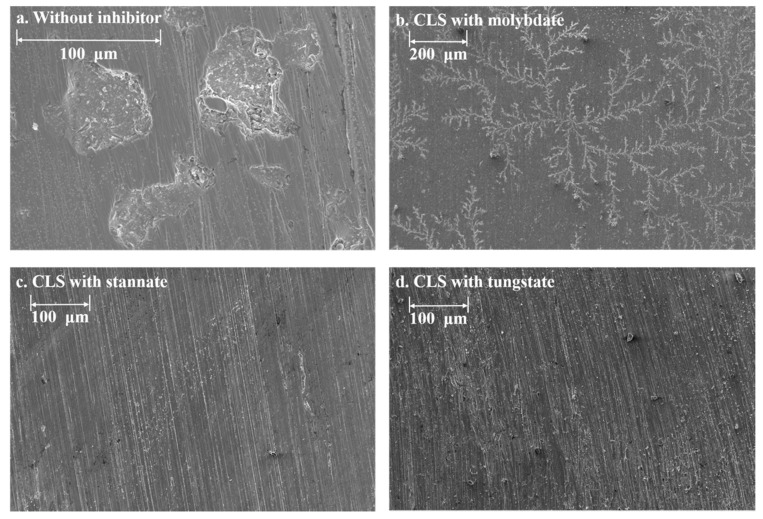
The surface morphology of Q235 carbon steel immersed in test solution without or with 500 ppm hybrid inhibitors: (**a**) without inhibitor; (**b**) CLS with molybdate; (**c**) CLS with stannate; (**d**) CLS with tungstate.

**Figure 10 molecules-25-04200-f010:**
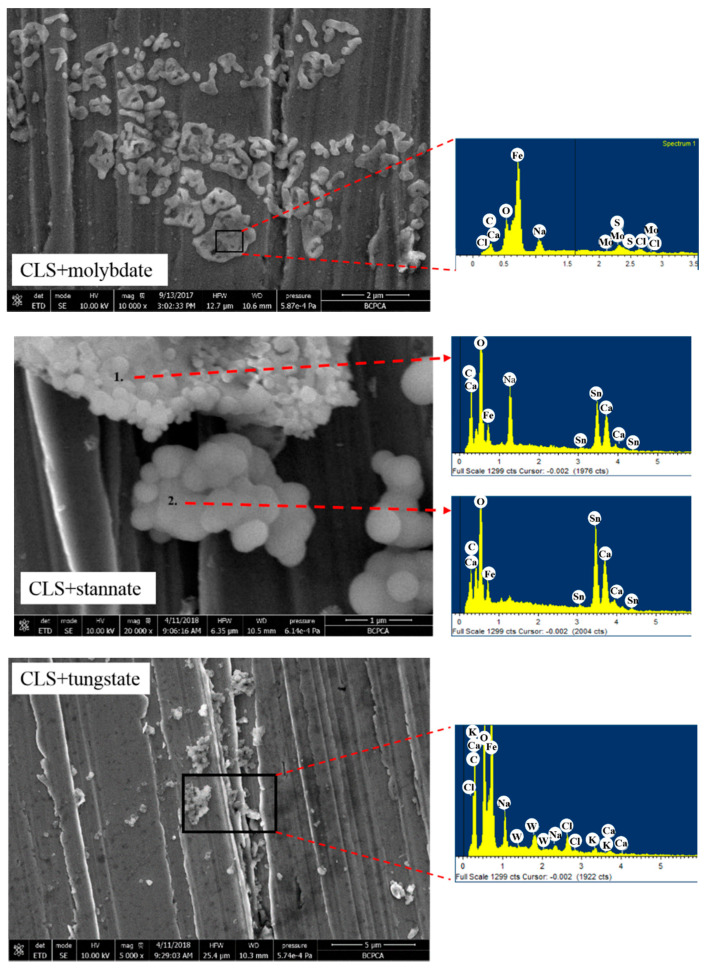
High resolution scanning electron microscope (SEM) images and energy dispersive X-ray spectrometer (EDS) results of the hybrid inhibitors treated steel surface.

**Figure 11 molecules-25-04200-f011:**
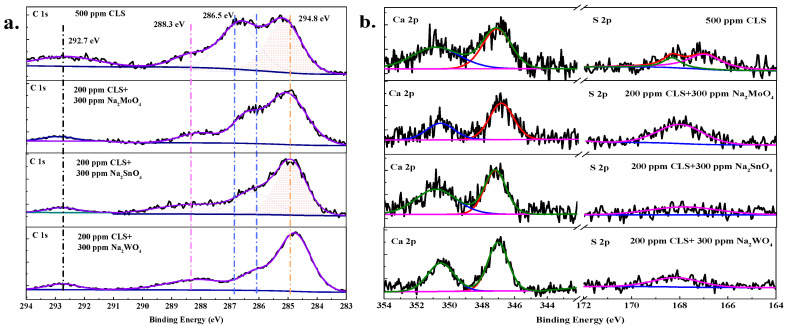
High-resolution XPS spectrum of Q235 carbon steel in the test solution with CLS and inorganic inhibitors: (**a**) C 1s; (**b**) Ca 2p and S 2p.

**Figure 12 molecules-25-04200-f012:**
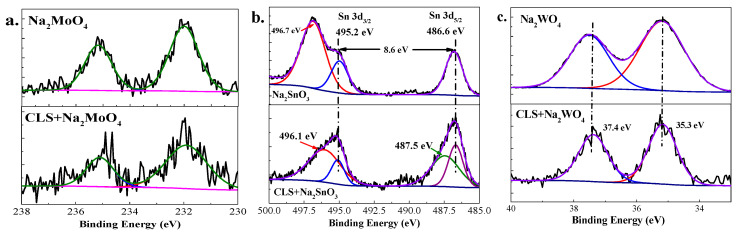
High-resolution X-ray photoelectron spectroscopy (XPS) spectrum of Mo 3d (**a**), Sn 3d (**b**), and W 4f (**c**) for inorganic inhibitor treated steel and CLS + inorganic inhibitor treated steel.

**Figure 13 molecules-25-04200-f013:**
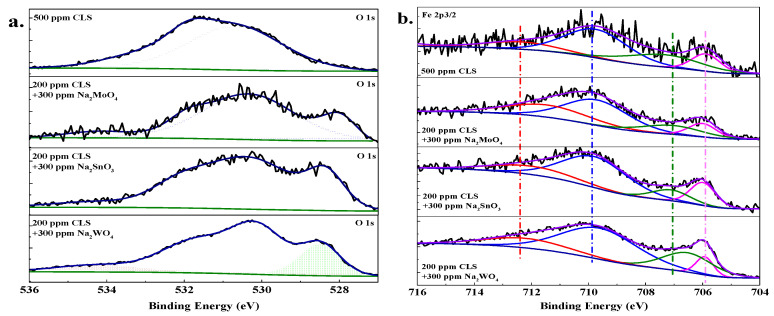
High-resolution XPS spectrum of Q235 carbon steel immersed in test solution with different inhibitors: (**a**) O 1s; (**b**) Fe 2p.

**Figure 14 molecules-25-04200-f014:**
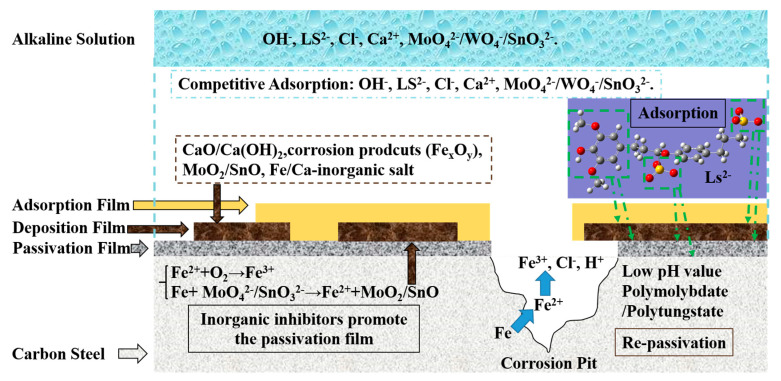
The schematic illustrations of the inhibition mechanism.

**Table 1 molecules-25-04200-t001:** The components of Fe in the surface film of inhibitor treated steel. CLS, calcium lignosulfonate.

Position	711.7 eV (FeOOH/Fe_2_O_3_)	709.7 eV (Fe_3_O_4_)	707.1 eV (Fe^2+^)	706.0 eV (Fe0)
CLS	14.49%	48.31%	24.15%	13.04%
CLS + Molybdate	28.57%	46.08%	14.29%	11.06%
CLS + Stannate	18.58%	54.64%	10.93%	15.85%
CLS + Tungstate	17.65%	53.48%	21.93%	6.95%

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
