# Peer review of "Study on Synergistic Corrosion Inhibition Effect between Calcium Lignosulfonate (CLS) and Inorganic Inhibitors on Q235 Carbon Steel in Alkaline Environment with Cl−"

_molecules, 2020, doi:10.3390/molecules25184200_

Round 1
Reviewer 1 Report
The results are interesting and they are reported in a clear way. I only have a few minor comments:
1) There is a lack of information how many samples were prepared for each corrosion medium? Two, three or more?
2) Why Q235 carbon steel was used in this study?
3) I guess the tests were repaeted more than one time, if so, what was the standard deviation or scatter or error bars of the results presented in Fig. 2, 4 and 7?
4) I suggest moving chapter 4 as chapter 2. Then chapter 3 would be "Results" and chapter 4: ”Discussion”.
5) It would be also worth clarifying where the results could be applied. Do the results have only cognitive or also applicable character?
Author Response
Dear Reviewer 1:
Thank you so much for your time and suggestions to our manuscript entitled “Study on synergistic corrosion inhibition effect between Calcium lignosulfonate (CLS) and inorganic inhibitors on Q235 carbon steel in alkaline environment with Cl-” (Molecules-919297). These comments are all valuable and very helpful for revising and improving our paper, as well as the important guiding significance to our researches. We have studied your comments carefully, and revised this manuscript with much attention to your comments and suggestions. And the answers to your comments are as follows:
1) There is a lack of information how many samples were prepared for each corrosion medium? Two, three or more?
Thank you for suggest. In order to ensure the experimental results can be reproduced, all electrochemical tests have 4 times parallel experiments. And the parallel experiments times have been added into the article.
2) Why Q235 carbon steel was used in this study?
Thank you for your question. The corrosion medium tested in this research is an alkaline environment, which is a common corrosion condition in reinforced concrete system, marine environment and so on. Carbon steel has a lot of applications in this corrosion environment, and the corrosion inhibition method of carbon steel in such environment becomes a necessary research subject.
3) I guess the tests were repaeted more than one time, if so, what was the standard deviation or scatter or error bars of the results presented in Fig. 2, 4 and 7?
Thank you for suggest, we have added the error bars of the results based on parallel tests and presented in Fig. 2, 4 and 7.
4) I suggest moving chapter 4 as chapter 2. Then chapter 3 would be "Results" and chapter 4: “Discussion”.
Thank you for suggest, we have looked up several articles published in Molecules, and the used article structure is the most common one used in Molecules.
5) It would be also worth clarifying where the results could be applied. Do the results have only cognitive or also applicable character?
Thank you for your question. In the first place, the research results could help us further understanding the corrosion behavior and corrosion inhibition methods of carbon steel in alkaline corrosion environment. Furthermore, we plan to study the corrosion inhibition effect of tested hybrid inhibitors on reinforced carbon steel rebars in concrete system, in order to test the applicable character of the hybrid corrosion inhibitor.
Reviewer 2 Report
The work of this manuscript is a continuation of a previous article by the authors. They present a study on synergistic corrosion inhibition effect between Calcium lignosulfonate (CLS) and inorganic inhibitors on Q235 carbon steel in alkaline environment with Cl- . in previous work they had studied the inihibion of the Q235 carbon steel by calcium lignosulfonate and sodium molybdate. the work was scientifically well conceived and executed and finally presented. For the last issue it would need some improvements. Here below some minor revisions by using the line number.
Line 96 . For the structure of CLS give a reference
Line 172. please give an explanation of the ratios used. It is based only on the experience of the autors or otherwise
Fig 4,7. I suggest to better present the labels for the x axis. The label "Inhibitor" is part of the label "Inorganic and I have to read Inorganic Inhibitor, or I have to read as "ratio Inhibitor? What means "without" ? without any inhibitors or if I follow the ratio (4:1, 3:2, 2:3, 1:) its' only CLS?
Fig 5. The label A, B and C in figure caption are missed in the plots. In Figure caption, A is in uppercase and the others in lowercase. To better read the plots I suggest that for x and y axes (Nyquist plots) put x106 & and next using as labels only 0.5, 1, 1.5...
Fig.8 Again A, B , C and D in figure caption are missed in the images. Expect for the A for the others it's very hard to read the scale. Please increase the quality, or draw the scale, in white, inside the images.
Line 565. You missed the figure number (14 I guess)
Author Response
Dear Reviewer 2:
Thank you so much for your time and suggestions to our manuscript entitled “Study on synergistic corrosion inhibition effect between Calcium lignosulfonate (CLS) and inorganic inhibitors on Q235 carbon steel in alkaline environment with Cl-” (Molecules-919297). These comments are all valuable and very helpful for revising and improving our paper, as well as the important guiding significance to our researches. We have studied your comments carefully, and revised this manuscript with much attention to your comments and suggestions. And the answers to your comments are as follows:
Line 96 . For the structure of CLS give a reference.
Thank you for suggest, we have add the references of CLS chemical structure.
Line 172. please give an explanation of the ratios used. It is based only on the experience of the autors or otherwise
Thank you for your question, the ratio of organic inhibitor and inorganic inhibitor in this article is basing on the following references and the authors’ experience.
References:
Lin, B.; Zuo, Y. Inhibition of Q235 carbon steel by calcium lignosulfonate and sodium molybdate in carbonated concrete pore solution. Molecules 2019, 24, 518.
Zhou, Y.; Zuo, Y.; Lin, B. The compounded inhibition of sodium molybdate and benzotriazole on pitting corrosion of Q235 steel in NaCl+NaHCO3 solution. Mat. Chem. Phys. 2017, 192, 86–93.
Mehdi J.; Roya O. Synergistic inhibition behavior of sodium tungstate and penicillin G as an eco-friendly inhibitor on pitting corrosion of 304 stainless steel in NaCl solution using Design of Experiment. J. Mol. Liq. 2019, 291, 111330.
Gao, Y.; Hu, J.; Zuo, J.; Liu, Q.; Zhang, H.; Dong, S.; Du, R..; Lin, J. Synergistic inhibition effect of sodium tungstate and hexamethylene tetramine on reinforcing steel corrosion. J. Electrochem. Soc. 2015, 162, C555-C562.
Fig 4,7. I suggest to better present the labels for the x axis. The label "Inhibitor" is part of the label "Inorganic and I have to read Inorganic Inhibitor, or I have to read as "ratio Inhibitor? What means "without" ? without any inhibitors or if I follow the ratio (4:1, 3:2, 2:3, 1:) its' only CLS?
Thank you for suggest, we have changed the labels for x axis of Fig. 4 and 7 to make the labels more clearly.
Fig 5. The label A, B and C in figure caption are missed in the plots. In Figure caption, A is in uppercase and the others in lowercase. To better read the plots I suggest that for x and y axes (Nyquist plots) put x106 & and next using as labels only 0.5, 1, 1.5...
Thank you for suggest, we have changed the x and y axes labels of all Nyquist plots in Fig. 5 as your suggest.
Fig.8 Again A, B , C and D in figure caption are missed in the images. Expect for the A for the others it's very hard to read the scale. Please increase the quality, or draw the scale, in white, inside the images.
Thank you for suggest, we have re-draw the scale in white inside Fig. 8.
Line 565. You missed the figure number (14 I guess)
Thank you for reminding, we have amended in the article.